# Peculiarities of *Plasmodium falciparum* Gene Regulation and Chromatin Structure

**DOI:** 10.3390/ijms22105168

**Published:** 2021-05-13

**Authors:** Maria Theresia Watzlowik, Sujaan Das, Markus Meissner, Gernot Längst

**Affiliations:** 1Department of Biochemistry, Genetics and Microbiology, Biochemistry III, University of Regensburg, Universitätsstr. 31, 93053 Regensburg, Germany; Maria-theresia.watzlowik@ur.de; 2Faculty of Veterinary Medicine, Chair of Experimental Parasitology, Ludwig-Maximilian-University, Munich, Lena-Christ-Str. 48, 82152 Martinsried-Planegg, Germany; Sujaan.Das@para.vetmed.uni-muenchen.de (S.D.); Markus.Meissner@para.vetmed.uni-muenchen.de (M.M.)

**Keywords:** chromatin, epigenetics, *Plasmodium falciparum*, nucleosome, nucleosome remodeling, transcription regulation, chromatin structure

## Abstract

The highly complex life cycle of the human malaria parasite, *Plasmodium falciparum*, is based on an orchestrated and tightly regulated gene expression program. In general, eukaryotic transcription regulation is determined by a combination of sequence-specific transcription factors binding to regulatory DNA elements and the packaging of DNA into chromatin as an additional layer. The accessibility of regulatory DNA elements is controlled by the nucleosome occupancy and changes of their positions by an active process called nucleosome remodeling. These epigenetic mechanisms are poorly explored in *P. falciparum.* The parasite genome is characterized by an extraordinarily high AT-content and the distinct architecture of functional elements, and chromatin-related proteins also exhibit high sequence divergence compared to other eukaryotes. Together with the distinct biochemical properties of nucleosomes, these features suggest substantial differences in chromatin-dependent regulation. Here, we highlight the peculiarities of epigenetic mechanisms in *P. falciparum*, addressing chromatin structure and dynamics with respect to their impact on transcriptional control. We focus on the specialized chromatin remodeling enzymes and discuss their essential function in *P. falciparum* gene regulation.

## 1. Introduction

*Plasmodium falciparum*, a unicellular eukaryotic parasite, causes the most severe and deadly form of the human disease malaria. In 2019, 229 million cases of malaria infection with about 409,000 deaths were reported, mainly affecting children under the age of five [1]. Malaria is still a major threat for humans and the situation may become worse as parasites increasingly develop resistance to the frontline choice of treatment, artimisinin-based combinational therapy, just as resistance to other effective drugs has emerged [2]. There is an urgent need for new antimalarial drugs, but this requires better understanding of the physiological, biochemical and pathological mechanisms of the parasite. In particular, the chromatin landscape and the epigenetic mechanisms are exceptionally different in *Plasmodium*, representing a potential drug target. Here, we review the specific features of *Plasmodium falciparum* chromatin structure.

*Plasmodium* spp.—together with other parasites such as *Toxoplasma*—belong to the phylum of Apicomplexa, which evolutionarily separated from the human line about 800–1000 million years ago [3]. The *Plasmodium* life cycle is complex, including two different hosts, with an asexual reproduction phase in humans and sexual reproduction in mosquitos of the genus *Anopheles* [4]. In humans, *Plasmodium* exists in intra- and extracellular forms and is capable of invading various cell types, including hepatocytes and erythrocytes. The parasite undergoes asexual reproduction and sexual commitment in the human host, leading to the differentiation of male and female gametocytes, which are taken up by a mosquito bite for the completion of sexual reproduction. This versatile multistage life cycle is tightly regulated with defined sets of proteins being concertedly expressed in specific phases of the life cycle. The underlying regulatory mechanisms of this highly orchestrated gene expression program in *Plasmodium falciparum* are poorly understood.

As a common principle, gene regulation in eukaryotes involves regulation on many levels, starting with controlling DNA accessibility within chromatin, followed by transcriptional control, post-transcriptional regulation, translational control, protein stability and activity, and additional mechanisms [5]. In this review, we will focus on *Plasmodium falciparum* chromatin structure and dynamics, reviewing its contribution to the regulation of gene expression and comparing it to other eukaryotes.

## 2. DNA-Based Features

### 2.1. Genome and Gene Architecture

The complete *Plasmodium falciparum* (*Pf*) genome sequence was determined in the year 2002 [6] and consists of 23.3 million base pairs, organized in 14 chromosomes plus 6 kb mitochondrial and 34 kb apicoplast DNA located in the respective cytoplasmic organelles [7]. The parasite genome features one of the most AT-rich eukaryotic genomes with an overall AT-content of 80.7% and within intergenic regions and introns of up to 95% [8]. Due to the high AT-content, the genome contains numerous low-complexity regions, simple sequence repeats and a skewed codon usage bias. It comprises a total number of 5280 protein-coding genes as well as 158 pseudogenes and 103 annotated noncoding RNAs [9]. At least 4557 of the 5280 genes are transcribed and expressed in a complex pattern depending on the life cycle stage [10], requiring a complex regulatory network.

In general, the *Pf* genome exhibits typical eukaryotic features, with genes consisting of exons and introns separated by intergenic regions. But gene architecture clearly differs from other unicellular eukaryotes by an increased mean gene length of 2300 bp (vs. 1400 bp in *Saccharomyces*), an increased mean exon length of 949 bp (typically 200–300 bp in all eukaryotes) and a markedly large proportion of genes larger than 4000 bp. The sizes of introns and intergenic region usually correlate linearly with the genome size [11]. With regard to genome size, *Pf* displays unusually large intergenic regions (mean of 1700 bp) and introns with an average length of 180 bp, which is rather long for protists but very short in comparison to higher eukaryotes [6].

### 2.2. Regulatory DNA Elements

Like other eukaryotes, *Pf* genes exhibit the characteristic bipartite structures of *cis*-regulatory regions with enhancer elements and basal promotors required for the recruitment of RNA polymerase II to the transcription start site (TSS) (reviewed in [12,13]). Most TSSs are relatively distant to the first exon, resulting in mRNAs with exceptionally long 5′ untranslated regions, with an average of 346 nt [14], when compared to human mRNA (~150 nt) [15]. The presence of antisense transcripts indicates the existence of bidirectional promotors with multiple shared or separate regulatory elements including the presence of multiple clusters of TSSs within a single gene locus [16]. Thus, genome-wide mapping of TSSs revealed highly diverse sets of start sites that are far more variable than those of human genes. Core promotors exhibit sequence motifs directing TSS selection and promotor strength, such as local changes in GC-content and homopolymeric nucleotide stretches (reviewed in [13]). Early on it was shown that the packaging of the promotor DNA into nucleosomes and alterations in nucleosome positioning and histone composition also influence gene activity [17,18,19], as will be described below.

Applying in silico approaches and algorithms, numerous putative *cis*-regulatory motifs could be predicted [20,21,22], and a few of them were experimentally validated [23]. These elements act as enhancers or silencers, as known for other eukaryotes, but *Pf* possesses an extraordinarily high number of such elements (4–5 per gene) sharing no sequence similarity with those of other eukaryotic organisms. Recent studies showed that the majority of the plasmodial genome is organized in regulatory units containing multiple genes and multiple regulatory elements with coordinated activity, rather than a one-on-one allocation of *cis*-regulator elements to their neighboring genes [24].

### 2.3. Trans–Acting Factors

The principle of gene expression regulation is based on the binding of specific transcription factors (TF) to *cis*-regulatory motifs. Surprisingly, only 73 TFs were identified in *Pf* having more than 5000 genes. This is far below the numbers in yeast (~170 TFs for 5400 genes) and human cells (>1500 TFs for 20,000 genes) [25,26,27]. In addition, most of the major families of eukaryotic TFs, such as homeodomains, basic leucine zipper, GATA fingers, nuclear hormone receptor and FKH domains, could not be identified through a homology search [28]. The *Pf* transcription factors can be grouped into eight helix-turn-helix proteins, 37 C2H2-type zinc fingers and one β-scaffold factor, but all exhibit only low conservation across different *Plasmodium* species [13,25]. Moreover, Apicomplexa possess a novel category of TFs, the ApiAP2 family, which are presumably the main regulators of transcription in the parasite life cycle [29,30].

Most of the 27 members of the ApiAP2 family were shown to exhibit sequence-specific DNA-binding [31,32]. Some of the *Pf*ApiAP2s or orthologs were shown to be essential and to drive transcriptional regulation at different stages of the life cycle [13], [33]: For example, AP2-L plays a critical role in liver-stage development [34]; AP2-G was identified to be the master regulator of gametocytogenesis [35]; AP2-I is relevant for invasion gene activation [36]; AP2-O activates gene expression in ookinetes [37], while gene expression in liver-infecting sporozoites is regulated by AP2-Sp [38]. Still the question remains, how such a low number of factors is sufficient to coordinate the complex gene expression profile of more than 5000 genes. It was suggested that TFs act in a combinatorial fashion and may exhibit pleiotropic functionality [33,39,40]. Alternatively, they might interact with additional regulatory proteins [36,41], or undergo post-transcriptional modification like other epigenetic regulators [42,43,44].

The interplay between *cis*-acting elements and *trans*-acting factors is strongly influenced by the packaging of the genomic DNA into chromatin. Nucleosome positioning and dynamics control the accessibility of the regulatory DNA elements for the *trans*-acting factors, as histones would mask the binding sites and inhibit DNA sequence recognition. Therefore, nucleosome-positioning plays an essential role, and well-defined chromatin architectures can be observed at regulatory regions. At gene promoters, positioned nucleosomes (+1) just downstream of the TSS can be observed, and the promoter region directly upstream is generally depleted of nucleosomes [18]. Such a structure is compatible with the binding of transcription factors, and changes to this chromatin structure would substantially affect gene activity. Genome-wide profiling using ATAC-seq identified such accessible DNA regions, mainly located in 5′-intergenic regions, overlapping with annotated and predicted *cis*-regulatory elements. These accessible regulatory regions correlate overall with high mRNA levels of the associated genes [45,46], revealing the binding of *trans*-activating factors to these sites. The direct effect of transcription-factor binding to transcriptional regulation was proven, but it is unclear how chromatin structure and nucleosome dynamics additionally affect and regulate the access of the *trans*-acting factors to their binding sites. Chromatin dynamics and nucleosome positioning may be a consequence of transcription-factor binding or represent a preceding event regulated by chromatin-remodeling enzymes that determine DNA accessibility.

## 3. Chromatin features

### 3.1. Pf Nucleosomes and Their Special Properties

Nucleosomes are the basic packaging unit of chromatin, consisting of a histone octamer associated with 147 base pairs of DNA wrapped around the proteins in 1.65 turns. The octamer consists typically of the four canonical histones H2A, H2B, H3 and H4, which are—due to their central function in DNA packaging—highly conserved in sequence throughout eukaryotic evolution [47]. For example, *Arabidopsis thaliana* H3 and human H3 differ in only two amino acids [48]. However, *Pf* histones show an extraordinarily high divergence with sequence identities of only 64%, 68%, 93% and 92% between human and *Pf* H2A, H2B, H3 and H4 respectively. The fifth histone H1, usually linking nucleosomes and promoting higher order structure, is not present in *Pf* [6].

In accordance with the diverging sequence of histones, plasmodial nucleosomes exhibit distinct biochemical properties when compared to human nucleosomes (see Figure 1). The *Pf* nucleosome exhibits reduced stability, weaker binding of H2A and H2B and has intriguingly lost the capability of DNA sequence-dependent nucleosome positioning [49].

Since DNA is not a flexible polymer but a rather rigid molecule with a persistence length of 150 bp, a deviation from this defined DNA structure would require bending energy [50]. In the context of a nucleosome, the DNA molecule has to be highly bent in order to enable wrapping around the histone octamer. The required bending energy is compensated by establishing about 400 direct and indirect ionic interactions and H-bridges between the DNA and the histones [51]. As GC and AT base pairs do not have exactly the same size and geometry, sequence composition does affect DNA structure by inducing bents and kinks with specific sequence motifs. Repetition of such motifs every 10 bp would induce directed DNA curvature that mimics the folding around the histone octamer, requiring less bending energy for nucleosome formation and thus, representing the preferential binding sites of nucleosomes [52]. It is a general eukaryotic principle that the genome sequence codes for a basic chromatin architecture with the base-pair sequence favoring nucleosome positioning and occupancy at specific sites [53,54,55]. In vitro and in vivo studies showed that nucleosome positions are in part determined by intrinsic DNA features [54]. The GC-content and the frequency of polyA-stretches and certain dinucleotide repeats turned out to be critical determinants for nucleosome positioning and occupancy and, therefore, impact the overall regulation of gene expression [56].

However, the AT-rich plasmodial genome sequence deviates substantially from other eukaryotes with respect to sequence motifs, and likewise *Pf* nucleosomes do not obey the classical sequence-dependent positioning rules [49]. The typical 10 bp periodicity of anti-phased A/T and G/C dinucleotides in nucleosomal DNA [57] is only weakly detectable in *P. falciparum* [18], [58]. The shifted nucleotide-ratio in the *Pf* genome creates a very different basis for chromatin structure, and *Pf* histones show deviating affinities when forming nucleosomes. Nucleosome positioning analysis in *Pf* revealed a significant number of positioned nucleosomes in vivo, mainly located at or in the vicinity of regulatory regions, as expected for their regulatory role in determining DNA accessibility. This raises the question of the mechanisms being responsible for nucleosome positioning in vivo, even though the *Pf* histone octamer does not recognize the underlying sequence code. There are numerous mechanisms that can still contribute to nucleosome positioning in *Pf*, including DNA binding factors, statistical positioning of neighboring nucleosomes by a constant DNA linker length and chromatin-remodeling factors that move and position nucleosomes [59]. For *Pf* nucleosomes it was shown that AT-repeat sequences in the DNA linker regions are sufficient to position nucleosomes. This presents a novel signal and mechanism for nucleosome positioning and suggests that histone linkers may interact with the linker DNA, thereby contributing to nucleosome positioning. Crystal structures of *Pf* nucleosomes are not available so far, but in silico modeling based on human nucleosomes suggests an overall similar nucleosome core structure with only a few divergent amino acids in the histone–DNA interacting regions [49]. The majority of sequence variation is located in the flexible histone tail regions, whose likely contribution to nucleosome positioning has not been proven yet. The fact is that *Pf* nucleosomes show decreased nucleosome stability overall, with weaker binding of H2A/H2B-dimer within the octamer, and an attenuated histone–DNA interaction increases the mobility of nucleosomes on DNA. The hypothesis that altered nucleosome properties evolved as an adaptation to the AT-rich plasmodial genome could be rejected. In vitro and in vivo data show that *Pf* histones like other eukaryotic histones bind preferentially to GC-rich over AT-rich DNA [49].

Besides the differences in nucleosomal properties, the nucleosome repeat length is highly divergent as well when compared to other eukaryotes. With a repeat length of 155 bp, the spacing of the nucleosome cores is maintained by DNA linkers with only 8 bp. Short nucleosome spacing is an intrinsic biochemical property of the *Pf* nucleosome, which can be observed in vitro and is maintained also in vivo [49,60]. The mean linker length in eukaryotes is significantly longer, varying from 20 to 75 bp depending on species and cell type [61]. Such short linker lengths, as found in *Plasmodium*, were shown to inhibit the folding of the nucleosome array into compact higher order structures of chromatin. The absence of higher-order chromatin structure correlates well with the assumption that *Pf* chromatin exhibits high accessibility [49,62,63,64]. The extremely short linker length and unusual chromatin compaction are putative consequences of the exceptional nature of DNA composition and histone properties in *Pf*.

### 3.2. Histone Variation

In addition to canonical histones, eukaryotes express histone variants throughout the cell cycle, which differ in amino acid sequence. Sequence variation occurs predominantly in the (N-)terminal histone tails and leads to novel and different sites of post-translational modifications, potentially impacting their function and interaction with chromatin-modifying enzymes [65]. Among eukaryotic species, different sets of variant histones are prevalent, with some ubiquitous variants having specialized functions in DNA repair (H2A.X), transcription activation (H2A.Z), kinetochore formation (CenH3) and transcription in general (H3.3) [66].

In *P. falciparum*, a homologue of the universally present H2A.Z was identified, but no H2A.X. Surprisingly, Apicomplexa additionally express an unusual H2B variant histone, called H2B.Z, whose function is still unclear. Genome-wide profiling revealed similar binding sites of *Pf*H2A.Z and *Pf*H2B.Z and coimmunoprecipitation experiments confirmed the existence of nucleosomes containing both H2A.Z and H2B.Z in the same octamer [67,68]. This observation is shared with studies in other Apicomplexa, including *Toxoplasma gondii*, indicating a role in the regulation of gene expression (reviewed in [69]).

Histones H3.3 and CenH3, the two universal variants replacing histone H3, are present in *P. falciparum*, although they have not been characterized in detail. Histone H3.3 has eight amino acids substitutions compared to canonical H3 (one amino acid exchanged in human H3.3) and is believed to preferentially bind GC-rich repetitive regions, independent of transcriptional activity, potentially contributing to the regulation of *var* gene expression and immune evasion [70]. The second H3 variant, CenH3, is enriched at AT-rich sequences of the centromere and is implicated in chromosome segregation [70,71].

Histones are the target for post-translational modifications (PTMs), mainly comprising acetylation, methylation and phosphorylation of the histone amino termini, which alter histone properties and their interactions with DNA and chromatin proteins, affecting the functionality of the underlying DNA. The combinatorial nature and functional impact of these epigenetic modifications are defined as the “histones code” amplifying the information content and plasticity of chromatin with respect to the regulation of all DNA-dependent processes [72,73]. Here too, *P. falciparum* exhibits striking differences to other eukaryotes with an unusually large proportion of constitutively acetylated histones and a high number (500) of identified PTMs, including several novel modifications specific to *Plasmodium* or Apicomplexa [72,74,75]. A recent study showed, that a few universally eukaryotic PTMs, initially proposed to be absent in *P. falciparum*, are tightly regulated, and their presence is limited to specific life cycle stages [76]. The presence of some dynamic PTMs peaking in particular stages, such as H3K4 and H3K27 modifications, emphasize their contribution to gene-expression regulation throughout stage development [72,76,77].

### 3.3. Nucleosome Occupancy and Dynamics In Vivo

Nucleosomal distribution on DNA is described by the terms “nucleosome occupancy” and “nucleosome positioning”: Occupancy describes the probability with which a certain base pair is covered by a nucleosome, while nucleosome positioning is a measure of the probability of a given base pair to serve as start, dyad or end position of a nucleosome [78].

The reduced stability and loss of sequence-dependent positioning of plasmodial nucleosomes in vitro is reflected by the genome-wide analysis of chromatin structure in vivo. In *Pf*, large genomic regions lack positioned nucleosomes or even appear to lack histone octamers on DNA at all. Several studies have addressed the nucleosomal landscape in *P. falciparum*, using Sonication-ChIP, MNase-ChIP, MNase-Seq and other methods, such as ATAC-seq and FAIRE, to detect nucleosome-free regions [18,45,46,58,79,80,81]. In summary, these studies show higher nucleosome density in heterochromatin, but contradicting experimental results were obtained regarding nucleosome occupancy in genic or intergenic regions. This review focuses on describing the nucleosomal landscape. For this, we largely exclude the description of FAIRE- and ATAC-seq experiments, as these methods monitor accessible DNA regions, not nucleosomal architecture. Recent studies addressing human and *Drosophila* chromatin have revealed the existence of nucleosomes with different stability [64,82] and, in order to assess all nucleosomes on DNA, appropriate protocols have been established. It was shown that partial MNase digestion of chromatin with still intact di- and tri-nucleosome fragments improves the overall sequencing coverage of nucleosomal DNA and avoids the loss of MNase-sensitive nucleosomes that are preferentially located at regulatory regions [64,82]. Taking into account these recent insights, we focus on experimental data that omit MNase digestion bias, in order to provide a clearer overview of *Plasmodium falciparum* chromatin structure. The only study so far using limited MNase digestion conditions was performed by Kensche and colleagues [18].

Intriguingly, the transcriptional unit of a typical gene in *P. falciparum* is framed by positioned nucleosomes upstream and downstream of the coding region (see Figure 2), resulting in the covering of regulatory regions and functional elements in the genome by positioned nucleosomes. A positioned +1 nucleosome can be mapped right at the TSS next to an upstream nucleosome-depleted region (NDR) of variable size and a detectable -1 nucleosome upstream of the NDR. This is a common pattern in eukaryotes, albeit the clarity and effectiveness of nucleosome positioning at these sites appear to be relaxed. The width of the NDR varies between individual *Pf* core promotors with a tendency of larger NDRs being associated with higher transcription levels. Positioned nucleosomes can also be detected at start and stop codons, as well as at splicing sites, which are relatively static throughout the life cycle. Nucleosomes positioned at start/stop codons may occur solely because of the increased GC-content of coding sequences, whereas those at exon–intron boundaries might be actively positioned to allow recruitment of post-transcriptional machineries. Overall, these observations suggest that nucleosome positions somehow highlight transcriptionally relevant landmarks, but positioning is less stringent and more fuzzy when compared to other eukaryotes. The MNase-seq data also indicated the typical 10 bp periodicity signal for AA/TT-dinucleotide driven nucleosome positioning in genic and intergenic regions, suggesting it is an additional, albeit less pronounced, feature of nucleosome positioning in *Plasmodium*. Kensche and colleagues suggest this much fuzzier pattern to be a global effect originating in the merging of multiple genes displaying different nucleosome patterns. Alternatively, the fuzziness could be a local consequence of the divergent stability and positioning properties of *Pf* nucleosomes within individual genes and thus, point to a distinct chromatin organization [49].

Comparative analysis of nucleosome-positioning dynamics at different life cycle stages shows that most nucleosomes in the transcriptional unit are static and nondynamic, indicating no gross changes of chromatin structure with variable gene expression. However, upstream promotor regions show significant changes in nucleosome occupancy levels during the life cycle, correlating with changes in gene transcription. With increasing transcriptional activity, nucleosome-depleted regions appear, which may be related to the formation of the RNA polymerase II initiation complexes. Accordingly, gene repression correlates with dynamic increases in nucleosome levels inhibiting transcription initiation [18,58,80]. The data suggest local changes in nucleosome occupancy around specific DNA motifs within these 5′ intergenic region being indicative of transcription factor binding, whereas global chromatin structure stays unaltered throughout the life cycle [18].

Studies mapping the genomic localization of nucleosomes containing the histone variants H2A.Z and H2B.Z identified them in the intergenic regions of euchromatin domains, particularly enriched at gene promotors [19,67,68,69]. The variant nucleosome levels do not change during the life cycle, suggesting that they permanently mark promotors and regulatory regions. The transcriptional activation of the heterochromatic *var* genes represents an exception to this rule, as the H2A.Z/B.Z levels at these promoters correlate with increased transcription level [68]. Moreover, the histone variant *Pf*H3.3 is preferentially located at euchromatic coding and subtelomeric repetitive sequences unrelated to transcription, whereas in other eukaryotes H3.3 is incorporated at sites of active transcription [83]. Interestingly, *Pf*H3.3 incorporation was also found at promotors of poised and active (but not inactive) *var* genes pointing to its putative contribution to epigenetic memory in *var* gene expression [70].

Not only the histone variant distribution, but also the occupancy and positioning of *Pf* nucleosomes in general do not quite follow known eukaryotic principles. The highly divergent underlying determinants—DNA and histone properties—seem to shape a very different chromatin landscape in *Plasmodium falciparum*, and potential novel mechanisms may have evolved to allow for the tightly regulated gene expression program in the parasite. 

### 3.4. Chromatin Density and Nuclear Organization

Progressing through the erythrocytic life cycle, major changes in nucleosome occupancy were reported, obtained via high MNase digestion and Hi-C techniques [58,79,84]: the ring stages exhibit high nucleosome occupancy, whereas a depletion of nucleosomes was observed in the trophozoite stage, potentially opening chromatin for transcription and DNA replication. In the following schizont stage, nucleosome occupancy increases again, correlating with chromatin compaction for merozoite differentiation and egress. The chromatin structure in gametocytes was found to be relatively open. These global changes in chromatin accessibility within the life cycle are specific to *P. falciparum* and have not been reported to a comparable extent in any other eukaryote. 

Nuclear organization is proposed to be another epigenetic layer contributing to gene expression regulation using mechanisms such as rearrangement of chromosomes, locus repositioning and heterochromatic silencing [85]. Various studies have attempted to unveil the three-dimensional nuclear organization of *Plasmodium falciparum* using chromosome conformation capture techniques (reviewed in [85,86]). One characteristic of *P. falciparum* is the absence of chromosome territories supporting the presence of a relatively accessible chromatin structure. Although clustering for certain genomic domains, such as heterochromatin foci, telomeres and ribosomal DNA, could be shown, no pronounced chromosome condensation comparable to other eukaryotes was observed [87].

## 4. Epigenetic Regulation of Transcriptional Activity

As described in the previous sections, gene transcription is impacted by a combination of epigenetic features shaping the chromatin landscape: the main determinants are the variations of nucleosome occupancy and nucleosome positioning at specific DNA elements and histone variants, in addition to the dynamic histone PTMs as well as higher-order chromatin structures. Furthermore, long noncoding RNAs [88,89] have been suggested to contribute to gene regulation by serving as modular scaffolds and targeting modules that recruit chromatin-modifying enzymes to specific loci [90,91]. Finally, the driving force is the accessibility of promotors and enhancers within chromatin—implemented by correct nucleosome positioning—for transcription-factor binding and the initiation of complex formation [92].

Genome-wide MNase-seq [18] and ATAC-seq data [46] confirm this principle in *P. falciparum*: Kensche and colleagues identified 4821 dynamic nucleosomes with 80 percent being located in euchromatic intergenic regions, mainly at promotors. The dynamics of these nucleosomes clearly correlate with temporal transcriptional activity of the downstream gene. Toenhake and colleagues identified 4035 accessible regions, whereof 90 percent are located in intergenic regions. The majority was found to be associated with one or more putative promotors and to correlate in accessibility score with abundance of the downstream gene product.

For some genes with high transcriptional variation, the epigenetic mechanisms were investigated in more detail (reviewed in [85]): the group of invasion genes was found to exist in an activated state caused by an interplay of the transcription factor AP2-I with the bromodomain-binding protein BDP1 binding to H3K9ac [93]. A repressed state in contrast—extensively studied on the example of *var* genes—is maintained by heterochromatin protein HP1, histone deacetylase HDA2 and the histone lysine methyltransferase SET2 and marked by H3K9me3 rendering these genes heterochromatic [94,95,96,97]. Furthermore, a chromatin-remodeling enzyme [PF3D7_0624600] and sirtuin proteins influence chromatin condensation [98,99], and the incorporation of noncoding RNAs complement the *var* gene-switching mechanism by silencing the gene locus via its sense—activating via its antisense—lncRNA [90,100]. Sexual commitment is known to be regulated by the epigenetic cascade starting with Ap2-G expression being repressed by HP1, which is evicted upon gametocyte development 1 (GDV1) association to heterochromatin, and GDV1 itself is controlled by the *gdv1* antisense RNA [35,97,101,102].

These examples illustrate the multilayered nature of epigenetic regulation, but concomitant nucleosome occupancy and positioning were hardly taken into consideration. However, in all of the described processes, nucleosome occupancy determines any interactions with the underlying DNA locus and certainly contributes to the unveiled regulatory mechanism. The importance of nucleosome occupancy was highlighted in a machine-learning model, wherein the relevance of individual epigenetic features in relation to the entirety of transcription regulation was assessed [77]: a collection of genomic and epigenomic data sets including information about transcription factor binding motifs, patterns of covalent histone modifications, nucleosome occupancy, GC content and global 3D genome architecture were analyzed for their prevalence in high-/low-expression genes. This kind of comparative analysis emphasizes the relevance of histone modifications, nucleosome occupancy and 3D chromatin architecture and suggests transcription-factor binding to be less important for transcription regulation. 

## 5. Chromatin Remodeling Enzymes

All DNA-dependent processes require dynamic changes in chromatin organization to exert their DNA specific activities. For this, eukaryotic cells have developed numerous enzymes that change the organization of DNA packaging [103]. Chromatin-remodeling enzymes alter chromatin structure by moving nucleosomes, while chromatin modifiers leave their chemical marks on chromatin to change the physicochemical properties of the chromatin fiber or to target protein/RNA complexes to specific genomic loci. The high variability of global and local chromatin packaging states and the numerous chromatin modifications associated with different functional processes, demonstrate the superordinate role of chromatin proteins in eukaryotic cells. On that account, it is important to study this emerging field in *Plasmodium falciparum*, as the fundamental differences between chromatin proteins in humans and *Plasmodium* will provide new insights into the evolution and mechanisms of chromatin dynamics and may reveal new therapeutic options.

In a comparative genomics study, the evolution of transcription factors, chromatin-modifying and -remodeling enzymes in parasitic protists was reconstructed [104]: Intriguingly, chromatin proteins evolved over millions of years in independent eukaryotic lineages through the proliferation of paralogous families and acquisition of novel domain architectures, leading to an enormous variety and to highly diverse sets of enzymes. Some chromatin-modifying enzymes have been identified in *Plasmodium falciparum* (detailed review in [105]). In this review we will focus on the evolution and mechanisms of chromatin-remodeling enzymes.

Remodeling enzymes and the large multiprotein complexes they form exert a direct ATP-dependent effect on nucleosomes. The enzymes alter histone–DNA interactions, resulting in the eviction, exchange and assembly of individual histones or histone octamers, changing the structure and stability of nucleosomes or the movement of histone octamers on DNA to reposition nucleosomes (reviewed in [43,106]). In order to disrupt the very stable interaction between histones and DNA, these enzymes couple their activity to ATP hydrolysis. A highly conserved ATPase module is conserved within all chromatin-remodeling enzymes, which is split into a Snf2_N and a helicase C domain, separated by a P-loop. The enzymes generally exhibit several additional protein domains, determining their specificity in substrate recognition and interaction with other proteins or RNA to form a variety of protein complexes [107].

The essential molecular function of this enzyme family is the movement of nucleosomes in order to provide accessibility to certain DNA regions. However, the exact regulatory mechanisms of this process—with respect to which nucleosomes are recognized to be moved and what is the target position—have not been completely uncovered. There is some evidence of “high affinity” and “low affinity” nucleosomes representing a putative mechanism or at least one aspect of defining reaction educts and products [108]. Therefore, multiple factors, such as the recognition of DNA sequences and structures, nucleosome composition and histone PTMs, play a crucial role. The specificity of remodeling machines depends furthermore on the central motor protein as well as the composition of all the associated complex subunits dramatically changing the responsiveness to various substrates and recruiting mechanisms. [43,106]. In human and mouse cells, it was estimated that over 1000 different chromatin-remodeling complexes with distinct functions may exist. Their cell-type-specific combination and dosage probably determines the cell-type-specific chromatin architecture, the gene-expression network and the responsiveness to specific signaling pathways in the cell [43,106,108,109,110]. To name an example, Snf2H—one of the 53 remodeling enzymes in humans—was biochemically purified in 18 different multiprotein complexes emphasizing the complexity of the “remodeling code” [111].

The functionally different subfamilies of SWI/SNF ATPases are grouped by the homology of the helicase region according to Flaus et al. (indicated by coloring in Figure 3) [112]: Starting with the Snf2-like group, the ISWI subfamily is mainly responsible for nucleosome repositioning, playing a role in nucleosome stabilization and higher-order structure [113,114]. In contrast, Snf2 enzymes have a more disruptive effect on nucleosomes, and Lsh proteins are associated with transcription silencing, cooperating with methyltransferases [115,116,117]. Chd proteins, comprising the Chd1, Mi-2 and Chd7 subfamilies, possess specific nucleosome remodeling activities and are characterized by their additional chromodomain. They are involved in processes like chromatin assembly nucleosome spacing [118,119] or function as regulators of gene expression in functionally distinct complexes [120,121]. The principal functions of the Swr1-like group—identifiable by the divided helicase domain and HSA domain—include histone eviction and the exchange of variants [122]. Rad54-like proteins seem to change DNA topology and alter nucleosomal accessibility [123], while Rad5/16 is involved in DNA repair pathways using its characteristic RING finger domain [124]. The SSO1652-like family does not directly alter nucleosome structure but is proposed to interact with transcription factors and is recruited to DNA lesions [115,116,117,125]. Last, a function for the distant group has not yet been determined. This selective overview indicates the functional diversity between the subfamilies of enzymes that depend on the highly conserved ATPase domain in combination with the additional complex subunits. The interplay of all these complexes in the cell organizes the nucleosomal landscape in a complex manner and, with this, the accessibility of regulatory DNA elements. Regulation of nucleosome positioning, keeping nucleosomes over regulatory sites—OFF state—or moving them next to the binding sites of regulatory factor—ON state configuration—likely determines local gene activity states. This mechanism can be paraphrased as “barcoding” the nucleosome landscape and highlights the essential role of remodeling enzymes in regulation and cellular differentiation [126].

Hidden Markov model (HMM) profile studies in *P. falciparum* revealed 10 genes encoding putative SWI2/SNF2 ATPases [12] and sharing similarities within the several subgroups, but with particularly low conservation when compared to other eukaryotes. According to evolutionary studies of Iyer and colleagues, remodeling enzymes had their origins in the bacteriophage replication system, and thereafter, a set of six enzymes with conserved domain architecture was suggested to be present in the last eukaryotic common ancestor (LECA) (Figure 3, left panel) [104]. During evolution, prior to the origin of kinetoplasts and then chromalveolates, new families of remodeling enzymes evolved with precursors of the Rad5/16 group, distant group, ALC1, Lsh and Etl1. Early on, the apicomplexan line evolved ten chromatin remodelers, whereof eight proteins could be allocated in the evolutionary model of Iyer and colleagues, based on homology search and domain architecture (Figure 3, mid panel). Higher eukaryotes developed a huge variety of SWI2/SNF2 ATPases starting from a few ancient types, e.g. leading to more than 50 in human cells (Figure 3, right panel) [112]. In comparison, *P. falciparum* possesses a limited set of 10 enzymes, only one per family, and therefore lacks the typical redundancy seen in higher eukaryotes, suggesting that these enzymes perform essential tasks in the cell. This fundamental difference in numbers between human cells and *P. falciparum* is accompanied in highly divergent domain architectures. Some domains such as PhD, Chromo and SANT domains, could be identified at very low stringency in sequence comparison but with no indication of their specific functions in *Pf*.

The individual remodeling enzymes in *P. falciparum*, and Apicomplexa in general, remain poorly characterized so far. In a pioneering study, Ji and Arnot identified and classified the first SWI2/SNF2 enzyme (Snf2L) with approximately 60% sequence homology to the ATPase domain of the yeast ISWI remodeler [127]. Since then, only a few studies addressed apicomplexan remodelers [98,128]. A genome-wide mutagenesis screen proposed one of the plasmodial remodelers to be essential and another two as putatively essential in asexual blood stages [9]. The authors suggest that the rest are not crucial for parasite fitness, not precluding the possibility of significant roles in chromatin organization. 

In general, perhaps with the exception of Chd1, all chromatin-remodeling enzymes are part of large multiprotein complexes with characteristic binding partners that link the enzymes to chromatin modifiers, chromatin and DNA binding motifs and alter the functionality of the complexes [109]. Interestingly, homology searches did not retrieve a single homologue of these proteins in *P. falciparum*. This astonishing lack of known interactors in combination with the reduced number and high divergence in sequence indicates that the chromatin-remodeling system in *Plasmodium* differs from known mechanisms in higher eukaryotes.

## 6. Potential Regulatory Network

As elaborated in the previous sections, *Plasmodium falciparum* chromatin is unusual in many respects, from the reduced number of transcription factors to high nucleosome dynamics and the divergent chromatin-remodeling enzymes/complexes. Therefore, we hypothesize that *P. falciparum* possesses a highly regulated but distinct chromatin system when compared to other eukaryotes. Here we address how these differences may affect the regulatory network and the transcriptional program throughout the parasite life cycle (Figure 4).

One crucial parameter for transcription regulation is the nucleosome landscape, which was investigated in several studies within the last few years. Conclusively, these studies point out a highly dynamic chromatin structure and atypical nucleosome features in *P. falciparum* [18,49,79]. The overall *Pf* chromatin was found to have a more open structure; the spacing of nucleosomes is surprisingly short; it is not organized in defined higher-order structures, and the nucleosomes themselves show reduced stability and lost capacity for sequence-dependent positioning (see Section 3.1 and Section 3.4). It is not known what exactly causes these altered nucleosome properties, but the extraordinary AT-rich genome and the extraordinarily high sequence divergence of *Pf* histones might interoperate to form this particular chromatin structure.

However, the applied methodology of chromatin analysis in *Plasmodium* parasites and the interpretation of the results is still a matter of discussion. On one hand, the AT-rich nature of the DNA is known to bias sequencing techniques [129]. On the other hand, the parasite rapidly moving through its highly different life cycle stages also blur the clarity of the observations. The fuzziness of *Pf* nucleosomal landscapes might be the consequence of highly unstable nucleosomes but could also represent the dynamic nature of chromatin in this organism. A strictly regulated but rapidly changing nucleosomal landscape is hard to capture precisely with the available techniques so far.

The second pillar of transcriptional regulation is formed by transcription factors and their interaction with *cis*-acting regulatory elements. The set of TFs that exist in *P. falciparum* was identified and characterized to some extent within the last few years, and the binding motifs and binding sites for many of them were identified. For the binding of certain transcription factors, such as Api2-G or Api2-O, defined downstream effects on the transcriptome were observed [32]. Since only a reduced number of TFs, primarily members of the ApiAP2 family were found, their relevance in gene-expression control is questionable. In any case, the mechanism of *trans*-regulatory factors is closely linked to the nucleosome landscape, since the accessibility of binding regions is an essential prerequisite for factor binding [92,130]. Therefore, promotor regions need to be nucleosome-depleted to enable TF binding. A detailed analysis of nucleosome positioning relative to TF binding regions would give us important information about the hierarchy of coaction between those two levels of regulation. Is the DNA region around the binding motif permanently nucleosome-depleted and thus accessible for TF-binding? If not, which mechanisms are responsible for uncovering these regions? Are some of the transcription factors pioneer factors that are capable of recognizing nucleosome-occupied motifs and subsequently recruiting chromatin-remodeling activities? These questions about specificity, recruitment and the order of events are important open issues in the field.

We propose that, for transcription regulation in *Plasmodium falciparum*, the chromatin-remodeling machinery is a third crucial determinant, strongly interlinked with the binding of transcription factors to DNA. As DNA properties explain the nucleosomal landscape only in part, active movement and positioning by remodelers has a high impact on nucleosome positioning and thus, on gene expression regulation [43,106,131]: remodeling complexes are recruited to target genes by transcription factors, RNA polymerases and elongation factors to promote or block transcription initiation and elongation by rearranging nucleosomes. They are the engines that both block (repression) or enable (activation) access to a gene through movement, positioning and the eviction/insertion of nucleosomes, depending on the specific type of remodeling enzyme at this locus. Remodeling complexes may barcode the genome in each stage of the life cycle, meaning that they establish a nucleosome-positioning landscape that allows the binding of certain factors and, conversely, restricts accessibility for other factors. Current investigations attempt to decipher this “remodeling code” and address the questions of how genomic loci are specifically recognized and how these enzymes are regulated. Which features determine the affinity to individual nucleosomes, serving as parameter for locus-specific nucleosome positioning? Another question is what is within the scope of function of the motor protein itself and what features are mediated by complex subunits? Looking at one level above, the regulation of remodeling enzymes themselves need to be investigated; it is presumed to occur in three different ways: control takes place (a) via recruiting to the correct target site by sensing the histone code or other factors [43]; (b) via adjustment of enzyme activity, e.g., by post-transcriptional protein-modification or ncRNAs [91]; or (c) by changing associated subunits and thus, conferring different activities to the complex (reviewed in [106]). 

How remodeling enzymes pave the way for transcription-factor and polymerase binding and how they are regulated is completely unexplored in *P. falciparum*. Based on the reduced number of identified *Pf* enzymes with high sequence divergence and the absence of any known interacting subunits, as there are in higher eukaryotes, we expect functional divergence. We propose that *Pf* remodeling complexes—closely linked with the nucleosome landscape and transcription-factor binding—build a complex regulatory network exhibiting major differences in comparison to other eukaryotes. The decryption of this system is indispensable to understanding the mechanism of transcription regulation in the parasite and will provide new insights and novel approaches for fighting malaria.

## Figures and Tables

**Figure 1 ijms-22-05168-f001:**
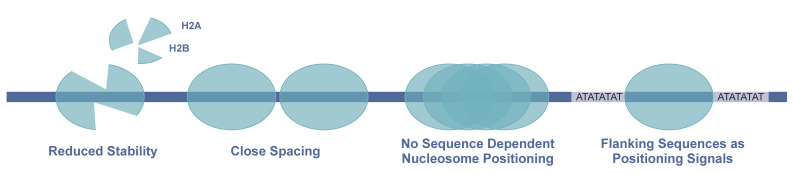
Schematic representation of distinct nucleosome properties in *P. falciparum*. In comparison with human nucleosomes, *Pf* nucleosomes exhibit reduced stability and short spacing between nucleosomes, and positioning is rather independent of internal DNA sequence but is determined by flanking sequence motifs.

**Figure 2 ijms-22-05168-f002:**
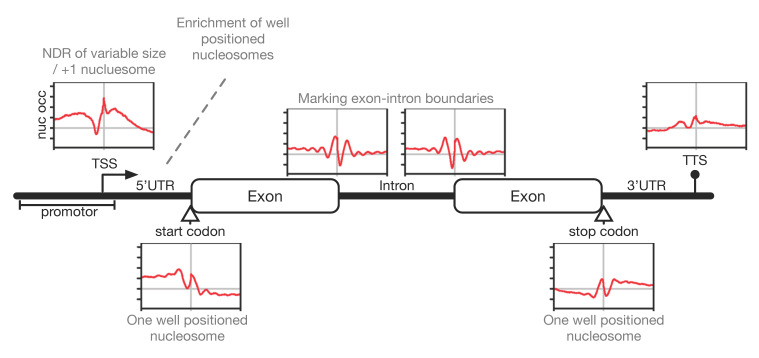
Schematic illustration of a typical gene with transcriptional landmark sites and corresponding plasmodial nucleosome positioning. The nucleosome profiles show the average MNase-seq occupancy normalized by gDNA aligned to the respective gene element combining eight time-points throughout the erythrocytic cycle according to [18]. Functional gene elements are designated, transcription start/stop is indicated arrow-/circle-tipped above the gene, translation start/stop codons are marked by arrows below the gene.

**Figure 3 ijms-22-05168-f003:**
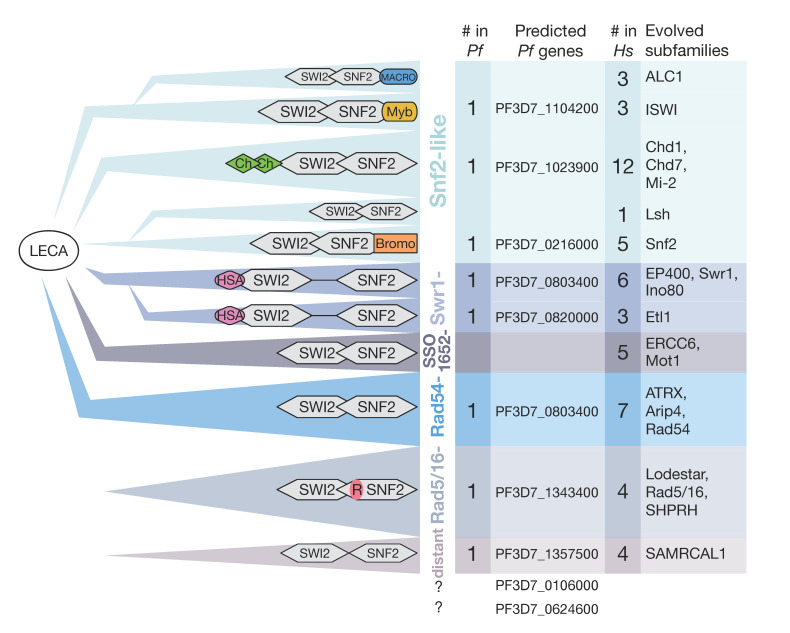
Evolutionary and phylogenetic development of SW2I/SNF2 ATPases. The evolution of SWI2/SNF2 starting from last eukaryotic common ancestor (LECA), according to [104], with triangles grouping together multiple subfamilies and their conserved domain architecture illustrated (left panel). Grouping of subfamilies and their names, according to [112], are indicated by coloring. SWI2-/SNF2-encoding genes in *P. falciparum*—wherever possible—were allocated to the categorization. Numbers of prevalent proteins for *P. falciparum* and *H. sapiens*, as well as names of further evolved subfamilies, are provided. Domain names: Ch = Chromo, R = RING, HAS = helicase/SANT-associated.

**Figure 4 ijms-22-05168-f004:**
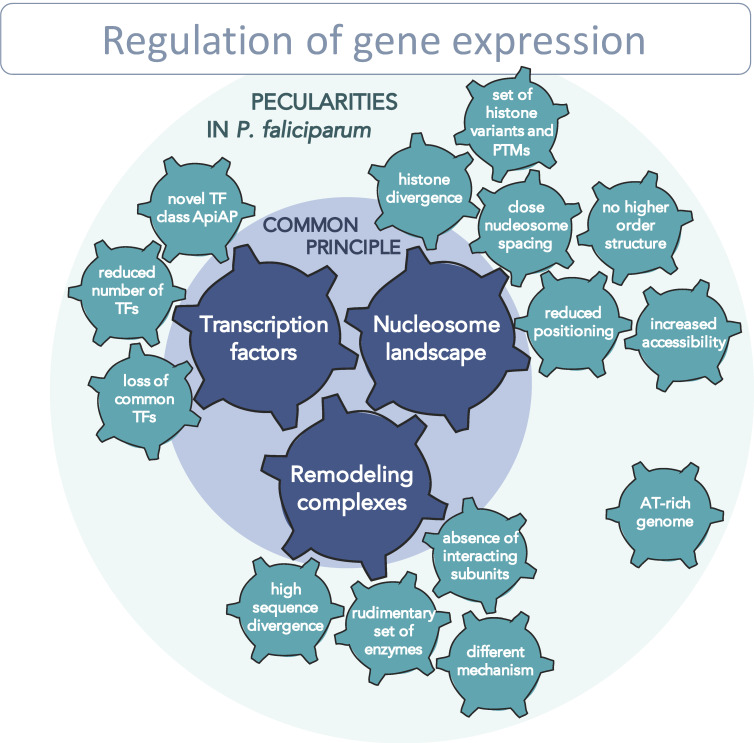
Schematic representation of the regulatory network of gene expression, with peculiar features in *P. falciparum* that affect the common principles.

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
