# Peer review of "Peculiarities of *Plasmodium falciparum* Gene Regulation and Chromatin Structure"

_ijms, 2021, doi:10.3390/ijms22105168_

Round 1

Reviewer 1 Report

In this review, Watzlowik et al. summarizes features of gene regulation mechanisms and chromatin structure of P. falciparum. This reviewer finds it very interesting and thinks that it merits publication with minor revision. Please find some suggestions for revision.

Major: 

  1. line 539: It says “no higher order structure could be shown”. Is this statement supported by previous reports? Is there no report of higher order structures such as physical interactions between genomic regions in P. falciparum analyzed by chromosome conformation capture or other methods? Please clarify.

Minor:

  1. The European style of writing numbers might be confusing. If it fits to the Journal style, please change it to that in other part of the world (e.g., line 30:409.000 to 409,000; line 61: 23,3 to 23.3, etc).
  2. line 451: please change “cell type specific” to “cell type-specific”.
  3. (iii)line 457: please correct error in “indicated by coloring in Error! Reference source not found.”
  4. (iv)line 589: please change “locus specific” to “locus-specific”.
  5. References section: many references are duplicated. Please correct the error. 

Reviewer 2 Report

Here, the authors present a very thorough description of the human malaria parasite Plasmodium falciparum with special emphasis on its genome composition and chromatin architecture/regulation. Overall, the work is extremely well-written and does a fantastic job of summarizing the knowns and unknowns from the current literature. Any newcomer to this area of chromatin research in this parasite should reference this review article. Aside from a few questions and minor comments below, this work is absolutely phenomenal.

  1. (Page-5) When the authors say that "nucleosomes have lower stability", does it mean that the octamer is less stable and there is more likelihood of observing hexamers, tetramers, and dimers? Or, does this imply that the octamer is simply less stably associated with the DNA itself (presumably because histone contacts with AT DNAs are weaker). Or both?
  2. (Page-7) Is there any speculation as to the importance of the nucleosome positions within the gene bodies (start/stop codons and exon-intron junctions) since they appear to static throughout the parasite's life cycle?
  3. (Page-10, line 457-458) There is a mistake here that should be corrected: 'Error! Reference source not found'.
